# RNA 2′-O-Methylation (Nm) Modification in Human Diseases

**DOI:** 10.3390/genes10020117

**Published:** 2019-02-05

**Authors:** Dilyana G. Dimitrova, Laure Teysset, Clément Carré

**Affiliations:** Sorbonne Université, Institut de Biologie Paris Seine, Centre National de la Recherche Scientifique, Transgenerational Epigenetics & Small RNA Biology, Laboratoire de Biologie du Développement, 75005 Paris, France; dilyana.dimitrova@sorbonne-universite.fr (D.G.D.); laure.teysset@sorbonne-universite.fr (L.T.)

**Keywords:** RNA modifications, 2′-O-methylation (Nm), human diseases, epitranscriptomics

## Abstract

Nm (2′-O-methylation) is one of the most common modifications in the RNA world. It has the potential to influence the RNA molecules in multiple ways, such as structure, stability, and interactions, and to play a role in various cellular processes from epigenetic gene regulation, through translation to self versus non-self recognition. Yet, building scientific knowledge on the Nm matter has been hampered for a long time by the challenges in detecting and mapping this modification. Today, with the latest advancements in the area, more and more Nm sites are discovered on RNAs (tRNA, rRNA, mRNA, and small non-coding RNA) and linked to normal or pathological conditions. This review aims to synthesize the Nm-associated human diseases known to date and to tackle potential indirect links to some other biological defects.

## 1. Introduction

### 1.1. What is Nm/2′-O-Methylation?

2′-O-methylation (or Nm, where N stands for any nucleotide) is a co- or post-transcriptional modification of RNA, where a methyl group (–CH_3_) is added to the 2′ hydroxyl (–OH) of the ribose moiety (Figure 1). 2′-O-methyl ribose can occur on any base and is an abundant and highly conserved modification found at multiple locations in transfer RNA (tRNA), ribosomal RNA (rRNA), and small nuclear RNA (snRNA) [1,2,3]. Nm is also present in a number of sites on messenger RNA (mRNA) [4] and at the 3′- end of small non-coding RNAs (sncRNAs), such as microRNAs (miRNAs) and small-interfering RNAs (siRNAs) in plants [4,5,6,7,8], on Ago2 loaded si- and miRNAs in flies, and on PIWI-interacting RNAs (piRNAs) in animals [9,10]. 

Nm has the potential to affect RNAs in multiple ways as it can increase their hydrophobicity, protect them from nuclease attacks [7,11], stabilize helical structures, and affect their interactions with proteins or other RNAs [12,13,14]. For example, Nm increases the thermodynamic stability of RNA:RNA base pairs and stabilizes A-form RNA duplexes [15,16,17,18]. The addition of Nm modification can also disrupt RNA tertiary structures [19] and inhibit RNA-protein interactions through steric effects [20,21] or by impacting hydrogen bonding [22,23].

### 1.2. Nm Detection

Detection of Nm modification has been a challenge for a long time. For example, classical biochemical strategies like liquid chromatography coupled with mass spectrometry (LC/MS) [24] and two-dimensional thin-layer chromatography (2D-TLC) [25,26] can be extremely effective, but time and labor consuming. Multiple other methods exploit the capacity of Nm to protect the adjacent nucleotide bond from cleavage or to impede the reverse transcriptase at low dNTP concentrations. Most of these techniques, as well as their improved variations, have to compromise between sensitivity, difficulty of execution (e.g., requirement for large amount of starting material and purification of individual RNA populations), and applicability to only specific populations of RNAs (high-abundance RNAs) [4,27,28,29,30,31,32,33]. RiboMethseq is a high throughput sequencing method for mapping Nm modifications that is based on the protection given by Nm against alkaline hydrolysis [34,35,36,37]. It seems to bring together most of the other techniques’ advantages. The main limitation of RiboMethseq is that it cannot be applied on short RNAs (<50nt) like sncRNA, for instance. Fortunately, a method based on the finding that Nm at the 3′ end of small RNAs can inhibit the activity of poly(A) polymerase was recently developed to detect such type of modification at this end position [38]. Although Nm-seq together with the RibOxi-seq methods [39] require bigger amounts of starting material than RiboMethSeq in general, they however all have the advantage of requiring a lower sequencing depth. For a recent comparison of sequencing-based Nm detection methods, see Reference [40].

It seems that we are finally well-equipped to detect and study Nm modification in a transcriptome-wide point of view. 

## 2. Nm in tRNA 

Transfer RNAs (tRNAs) are the most heavily modified RNA species and loss of certain modifications can cause deregulations in cellular processes and even diseases [41,42,43]. One of the most abundant modifications on them is Nm. Its function depends on the localization and as a consequence can be associated with general stability of the tRNA structure and/or proper translation [16,44,45,46].

### 2.1. *FTSJ1* Identity and Conservation

*FTSJ1* (also known as *MRX9*, *TRMT7*, *JM23*) is a characterized human tRNA 2′-O-methyltransferase that targets the C_32_ and N_34_ positions in the anticodon loop of tRNA^Phe^ (Figure 2) and tRNA^Trp^. It actually belongs to a large phylogenetically conserved superfamily of RNA methyltransferases, the RRMJ/fibrillarin superfamily [47,48]. *FTSJ1* homologs in *Escherichia coli* and *Saccharomyces cerevisiae* are respectively FTSJ/RRMJ and TRM7 (tRNA methyltransferase 7).

Expression of the human *FTSJ1* suppresses the severe growth defect of yeast *TRM7Δ* mutants [49]. In *S. cerevisiae*, TRM7 is the central component of a complex modification circuitry required for anticodon loop modification of target tRNAs. More concretely, TRM7 separately interacts with TRM732 and TRM734 partners to form Cm_32_ and Nm_34_ respectively, both of which are required on tRNA^Phe^ for efficient formation of a third modification, the wybutosine (yW) at m^1^G_37_ by other proteins [45,50,51]. Impressively, the same circuitry appears to be conserved in the phylogenetically distant yeast *Schizosaccharomyces pombe* [49]. This extremely conserved circuitry might be further extended in eukaryotes since suppression of the growth defect of *S. cerevisiae TRM7*Δ mutants by human *FTSJ1* expression requires the function of TRM732 or its human homolog THADA to form Cm_32_ on tRNA^Phe^ [49], and the formation of peroxywybutosine (o2yW_37_) at m^1^G_37_ is also affected in humans lacking *FTSJ1*. 

Interestingly, *FTSJ1* has two conserved paralogs: *FTSJ2* (mitochondrial rRNA methyltransferase 2 [Mrm2] in *S. cerevisiae*) and FTSJ3 (suppressor of poly-A binding protein Pab1p (Spb1) in *S. cerevisiae*), they however target different RNAs: 21S rRNA [52,53] and 25S rRNA [54,55], respectively. Finally, the bacteria ancestor of those three proteins FTSJ/RRMJ is an RNA-binding heat shock protein that targets the 23S ribosomal RNA [47]. *FTSJ2* and FTSJ3 are briefly discussed in the next chapters.

### 2.2. *FTSJ1* Link to Intellectual Disability

Intellectual disability (ID), or previously known as mental retardation (MR), is characterized by non-progressive cognitive impairment and affects 1–3% of the general population. X-linked genes seem to play a predominant role in ID as there are 10% more male than female ID patients cases reported [56]. One third of the X-linked ID (XLID) conditions are syndromic (SXLID) and the other two thirds are non-syndromic (NSXLID). As NSXLID has no obvious and consistent phenotypes other than mental retardation (IQ < 70), NSXLID conditions are clinically diverse and genetically heterogeneous disorders. 

*FTSJ1* gene is located on the small arm of chromosome X (Xp11.23), and accordingly, its loss of function has been identified as a cause for non-syndromic X-linked intellectual disability (NSXLID) [57,58,59,60,61,62,63,64]. Distinct alleles of *FTSJ1* from six independent families and one microdeletion affecting *FTSJ1* together with *SLC38A5* (solute carrier family 38 member 5) are linked to NSXLID (see Table 1 and updated from Reference [65]). Also, novel *FTSJ1* variations that appeared *de novo* in two other NSXLID patients are under investigation for further molecular details (Amélie Piton & Elise Schaefer Hôpitaux Universitaire de Strasbourg, personal communication and Ambry Genetics company reported in Clinvar). Heterozygous loss-of-function mutations in females do not cause the disease, which is most probably due to inactivation of the affected X chromosome [61,63].

Today, *FTSJ1* represents one of the strongest links between ID and a tRNA Nm modification gene. Consistent with the 2′-O-methyltransferase activity of *FTSJ1* on tRNAs, it was reported that tRNA^Phe^ from two genetically independent lymphoblastoid cell lines (LCL) of NSXLID patients (family 6 and A3—see Table 1) with loss-of-function *FTSJ1* mutations nearly completely lack Cm_32_ and Gm_34_ and have reduced peroxywybutosine (o2yW_37_) [64]. Additionally, tRNA^Phe^ from a patient carrying an *FTSJ1*-p.A26P missense allele (family 7, Table 1) specifically lacks Gm_34_, but has normal levels of Cm_32_ and o2yW_37_. tRNA^Phe^ from the corresponding *S. cerevisiae* trm7-A26P mutant also specifically lacks Gm_34_, and the reason is not a weaker TRM7/TRM73*4* interaction. These results strongly link defective Nm modification of the tRNA anticodon loop to the neurodevelopmental disorder observed in all patients that carry an *FTSJ1* mutation. It also points to the Gm_34_ of tRNA^Phe^ as a critical modification [64]. 

It is interesting to mention two more studies. First, three single-nucleotide polymorphisms (SNPs) in the *FTSJ1* gene showed a positive association with NSXLID [62]. Second, another analysis on the same three SNPs even suggests that genetic variations in *FTSJ1* are related to general cognitive ability, verbal comprehension, and perceptual organization in male individuals [67]. Although it is tempting to make a link between *FTSJ1* alleles and human cognitive ability, more profound studies are needed to support that idea.

### 2.3. Discussion on *FTSJ1*

Today, it is well established that mutations in the human tRNA 2′-O-methyltransferase *FTSJ1* cause NSXLID. *FTSJ1* is an S-adenosyl methionine (SAM) family methyltransferase and it is known that SAM has the ability to cross the blood-brain barrier [68]. SAM also serves as the sole methyl donor for methylation processes in the central nervous system [68], which makes the link between *FTSJ1* loss and a brain disease less surprising. So far, low cerebrospinal fluid levels of SAM have been observed in several neuropsychiatric and neurological disorders, including depression, brain ischemia, and dementia [68]. 

*FTSJ1* is generally widely expressed [60], though all mutations identified in NSXLID patients result in relatively “mild phenotypes” (lack of malformations or systemic breakdowns) that are essentially brain specific: delayed speech, concentration and learning difficulties, autistic behavior, motor tics, seizures, and others. It is possible that the activity of *FTSJ1* is most critical during brain development, which is supported by the relatively high expression of *FTSJ1* mRNA in the foetal brain [59]. It is also plausible that brain structures are more sensitive to defects in the translation machinery than are other organs, although the amount of mature tRNA, nor the charging of the concerned tRNAs does not appear to be affected in case of *FTSJ1* mutations [64] and the same is valid in *S. cerevisiae* [50]. However, and surprisingly, it has been shown recently by the same group that in *S. cerevisiae*, *TRM7*Δ mutants and the equivalent in *S. pombe*, the cells constitutively activated a robust general amino acid control (GAAC) response, acting through Gcn2 (general control nonderepressible 2), which senses uncharged tRNA [69]. Alternatively, another protein may partially compensate for the loss of FTSJ1, thereby preventing *FTSJ1* mutations from causing more severe disorders. As we will discuss later in this review, Nm enzymes seem to demonstrate certain flexibility when it comes to their targets. Thus, Nm enzymes, other than FTSJ1, may, in addition to their known targets, partially methylate positions C_32_ and N_34_ of tRNA^Phe^ and tRNA^Trp^. Finally, mRNA levels of Gcn2-dependent GAAC-regulated genes, *CTH* and *GADD153*, were not conclusively upregulated in cells carrying a mutation in *FTSJ1*.

Finally, *FTSJ1* function seems to be crucial for normal brain development, but the dosage control of its expression may be equally important. Some studies describe cases of ID in which the patients carry duplication in the Xp11.23 region. Those duplications include *FTSJ1*, but also other genes associated with ID [70,71]. That makes it impossible at the moment to conclude whether a supplementary dose of *FTSJ1* causes ID, the same way as the loss of *FTSJ1*. 

The association of *FTSJ1* with mental retardation highlights the importance of the Nm modification of RNAs, specifically in brain development and cognitive processes [65]. Current studies (including our group) aim to further characterize the mechanism of action of *FTSJ1* and the associated to its lack-of-function molecular phenotypes. Preliminary data of our group showed an accumulation of specific tRNA fragments (tRFs) in mutants of the *Drosophila* homolog of *FTSJ1*, which is in accordance with the nuclease protection that Nm usually confers to modified nucleotides (Angelova et al., in prep). We are currently investigating this molecular phenotype and its potential conservation in humans as it may have a role in the abnormal brain development in the *FTSJ1* patients.

### 2.4. TRMT44 and Idiopathic Epilepsy

TRMT44 is a putative 2′-O-methyluridine methyltransferase predicted to methylate residue 44 in tRNA^Ser^ [72]. Mutations in this gene were implicated in partial epilepsy with pericentral spikes, a novel mendelian idiopathic epilepsy. However, the underlying mechanisms that fails in those patients are currently unknown.

## 3. Nm in rRNAs

rRNAs are one of the most highly modified RNA species in the cell and definitely the ones that carry the most Nm modifications to date [34]. Those rRNA modifications are obtained at different stages of ribosome biogenesis and ensure efficiency and accuracy of the translation by stabilizing the rRNA scaffold [73,74]. There is also a certain heterogeneity in rRNA Nm modifications in disease that may alter ribosome function as a cause or as a consequence of the condition [75,76].

### 3.1. snoRNAs and Prader-Willi Syndrome

The Nm modification at each rRNA site is regulated by box C/D small nucleolar RNAs (snoRNAs) in association with fibrillarin methyltransferase (see next chapter). The snoRNA serves as a guide due to its sequence complementarity to the target rRNA and fibrillarin ensures the methylation reaction [77,78,79,80].

The essentialness of rRNA Nm modifications for vertebrate development has been exemplified in zebrafish. Loss-of-function of any of the following three snoRNAs: *SNORD26*, *SNORD44*, and *SNORD78*, leads to severe morphological defects and/ or embryonic lethality in this animal model [81].

Even stronger is the link between certain C/D box snoRNAs and Prader-Willi syndrome (PWS), a complex human neurological disease characterized with mental retardation, low height, obesity, and muscle hypotonia [82,83,84]. In several independent studies, it was shown that PWS is caused by the loss of imprinted snoRNAs in locus 15q11-q13. Large deletions of this region underlie about 70% of the PWS cases [85], whereas duplication of the same region is associated with autism [86,87,88]. Locus 15q11–q13 contains numerous copies of two C/D box snoRNAs: *SNORD115* (HBII-52) and SNORD116 (HBII-85) [89,90]. *SNORD115* is believed to play key roles in the fine-tuning of the serotonin receptor (5-HT2C) pre-mRNA [91,92], while *SNORD116* loss is thought to contribute in the etiology of the PWS [89,93,94].

### 3.2. Fibrillarin and Diseases: Identity and Conservation

Fibrillarin (FBL) is a 34 kDa nucleolar 2′-O-RNA methyltransferase and is located in the dense fibrillar component of the nucleolus to whom it owes its name. It associates with the U3 snoRNA, a C/D box family member and together they have been involved in the processing of precursor rRNAs [95].

Human FBL is 70% identical to yeast NOP1 and they are functional homologues since either human or Xenopus fibrillarin can complement (at least partially) a yeast *nop1* mutant. In yeast, Nop1 is an essential nucleolar protein that is associated with small nucleolar RNA and is required for ribosome biogenesis [96]. Fibrillarin-like proteins also occur in the domain of Archaea [97]. FBL’s function in ribosome biosynthesis appears to be maintained from the archaebacteria, which lack a nucleus, to *Giardia*, which contains a nucleus but lacks a prominent nucleolus, to higher mammals, which have both a nucleus and nucleolus [98,99]. This strong evolutionary conservation suggests a high importance for this enzyme in ribosome biogenesis.

Therefore, FBL is an essential nucleolar protein that catalyzes the 2′-O-methylation of ribosomal RNAs. In addition, FBL participates in pre-rRNA processing [95] and it was also reported that it could regulate rRNA transcription via the methylation of a glutamine residue of histone H2A in plants, yeast, and human [100,101]. It can be expected that changes in the level of FBL expression would seriously affect the translation process, but a direct confirmation of this hypothesis was received only recently [76]. The downregulation of FBL in human cells alters the intrinsic capacity of ribosomes to initiate translation from internal ribosome entry site (IRES) elements in a manner independent of translation initiation factors [76]. As a result, FBL can affect the course of some cellular processes. Indeed, FBL is required for early development in mice and zebrafish [102,103]. Fibrillarin mutant mice are non-viable due to a massive apoptosis in early embryos [102] and *FBL* gene depleted zebrafish are severely affected in the optic tectum and the eye due to impaired neural differentiation, and again, massive apoptosis [103]. FBL is also highly expressed in pluripotent embryonic stem cells of mice, where it regulates stem cell pluripotency [104].

Being implicated in such important biological processes, it comes as no surprise that FBL loss is associated with different disease conditions, some of which have been recently reviewed in Reference [105].

### 3.3. Fibrillarin and Cancer

Increased ribosome biogenesis is typically required in tumors in order to meet the high demand for protein production in the rapidly dividing cells [106,107]. As FBL plays a role in rRNA production, it is not surprising to find it overexpressed in different cancers [108,109,110]. It was demonstrated that FBL quantitative distribution is directly related to RNA polymerase I transcriptional activity (thus, to rRNA production), and is inversely related to cell doubling time in cancer cells [111]. More concretely, the knock-down of FBL in human prostate cancer lines suppresses proliferation and reduces clonogenic survival [108]. Also, FBL overexpression is associated with poor survival in patients with breast cancer and promotes cellular proliferation and resistance to chemotherapy of MCF7 breast cancer cells [109].

Interestingly, the tumor suppressor transcription factor p53 represses the expression of the *FBL* gene by directly binding to it. The suppression of p53 expression in human mammary epithelial cells leads to modifications in the rRNA methylation pattern at the single nucleotide level, impairment of translational fidelity, and increased IRES-dependent translation of key cancer-related genes, such as *IGF-1R*, *C-MYC*, *VEGF-A*, and *FGF1/2* [109]. On the other side, the oncogene transcription factor C-MYC binds to the 5′- upstream region of the *FBL* gene and stimulates its expression [108,112,113].

Curiously, a few older studies revealed that patients with hepatocellular carcinoma (HCC), gastrointestinal, lung, and ovarian cancers have autoantibodies to FBL [114,115,116]. One could imagine that their production is an attempt of the immune system to fight the tumor through eliminating a target that is advantageous for cancer cells. Even if those autoantibodies have also been found in systemic autoimmune diseases (see below), fibrillarin may still be an interesting anti-cancer target in drug design and therapies. Today, fibrillarin has not been exploited yet as an oncology target, but a very recent review emphasizes its potential as a therapeutic target that could lower the genotoxic effects of anti-cancer treatment [117].

### 3.4. Fibrillarin and Autoimmune Diseases

One of the most characteristic serologic features of systemic sclerosis (scleroderma, SSc), a collagen vascular autoimmune disease, is the occurrence of autoantibodies against nuclear and most notably against nucleolar antigens (antinucleolar antibodies (ANoA)) [118]. Multiple old studies have identified fibrillarin as the third major nucleolar scleroderma autoantigen [119,120,121,122,123,124] and have concluded that anti-fibrillarin is a marker for severe SSc [125]. Those ANoAs do not seem to affect the survival rate of patients with diffuse cutaneous systemic sclerosis, one of the most severe complications, and the events that initially trigger their production are not known [126,127]. However, and curiously, a case of a 22-year-old woman was successfully treated with long-term plasma exchange whose beneficial effect persisted for two years [128]. 

Anti-fibrillarin antibodies have been detected in a variety of connective tissue patients like the mixed connective tissue diseases (60%), CREST syndrome (calcinosis, Raynaud phenomenon, esophageal dysmotility, sclerodactyly, and telangiectasia syndrome, 58%), systemic lupus erythematosus (39%), rheumatoid arthritis (60%), and Sjogern’s syndrome (84%), in addition to SSc patients (58%) [129,130]. However, outside the connective tissue diseases spectrum, anti-fibrillarin antibodies are rarely seen.

Of course, in the case of autoimmune diseases, the development of autoantibodies against FBL is rather a consequence and not a cause of the pathology. Still, autoantibodies are expected to lead to a decrease in the protein levels of their target. It would be interesting to verify whether indeed FBL protein is less present in certain patient’s tissues, as well as whether its targets lack Nm modifications, and look for related/correlated effects in terms of symptoms and aggravation of the disease.

### 3.5. Fibrillarin: Possible Link to Other Diseases

Among FBL’s interaction partners is the survival of motor neurons (SMN) protein [131,132]. The *SMN* gene is the disease-causing gene of spinal muscular atrophy (SMA), a common and often fatal autosomal recessive disease, leading to progressive muscle wasting and paralysis as a result of degeneration of anterior horn cells of the spinal cord [133,134]. In human cell lines, the SMN protein localizes in gemini of Cajal bodies (gems), compartments frequently found near or associated with Cajal bodies (nonmembranous nuclear sub-organelles), in which FBL is found [135,136,137,138]. Among its other functions, SMN has been suspected to be involved in ribosome biogenesis [139], a cellular process in which FBL is strongly implicated. The exact role of SMN in SMA is not completely clear and it is unknown if its interaction with FBL is also involved. FBL immunofluorescence in Schwann cells revealed nucleolar segregation in a tellurium-induced demyelinating neuropathy rat experimental model [140]. It has not been excluded that FBL may play a role in that kind of disorder.

Finally, a recent study shows evidence of altered 2′-*O*-methylation of rRNA in 28S rRNA and increased levels of FBL expression in β-thalassemia trait carriers [141]. Today, thalassemia is the most important hematologic inherited genetic disease worldwide.

### 3.6. *FTSJ2* and Cancer

Another enzyme involved in rRNA Nm modification is FTSJ2, one of the two paralogues of *FTSJ1*. As already mentioned, *FTSJ2* is the human ortholog of the mitochondrial *E. coli* 23S rRNA 2′-O-methyltransferase (yeast’s Mrm2) [52,142]. *FTSJ2* has been implicated in cell proliferation and seen to be overexpressed in different cancer cell lines, especially in lung carcinoma cells [53]. Moreover, a later study found an amplification of the *FTSJ2* genomic locus, as well as an increase in *FTSJ2* mRNA levels in non-small cell lung cancer (NSCLC). Thus, *FTSJ2* has been identified as a new lung cancer oncogene candidate, even if its role in the disease in not completely clear today [143]. Surprisingly, another study compared two lung adenocarcinoma cell sublines and demonstrated that *FTSJ2* mRNA levels were lower in the more invasive one. In addition, an overexpression of *FTSJ2* in the highly aggressive rhabdomyosarcoma cells (TE671) turned out to inhibit their invasion and migration capabilities [142]. This seems to contradict what was just stated, but it is speculated that *FTSJ2* is advantageous for cancer cells only when they are in a less aggressive stage. *FTSJ2* expression may provide a transient advantage for the tumor, and once a certain threshold in the malignancy development is exceeded, *FTSJ2* may become a drawback and cancer cells try to eliminate it.

## 4. Nm in mRNAs

mRNAs can be co- or post-transcriptionally modified, which impacts their fine steady state level. When it comes to Nm modification, it has been mostly observed in the mRNA cap, but recently also internally in the coding DNA sequence (CDS). Indirect effects on mRNA expression can be observed when a lack of Nm affects the small nuclear RNA (snRNA)-regulated mRNA splicing.

### 4.1. mRNAs and Cap Protection

The cap structure is added to the 5′- end of eukaryotic and some viral mRNAs during transcription and is essential for mRNA stability, pre-mRNA processing, nuclear export, and cap-dependent protein synthesis [144,145,146,147,148,149]. The so-called “cap 0” (m7G(5′)ppp(5′)N) consists of a 7-methylguanosine addition to the first transcribed nucleotide (N1) [150]. Further 2′-O-methylation can be added on the N1 and often also on the N2 (second transcribed nucleotide); here we are talking about “cap 1” (m7GpppNmN-) and “cap 2” (m7GpppNmNm-), respectively (Figure 3). *CMTR1* 2′-O-methylates the N1 [151] and CMTR2 2′-O-methylates the N2 [152,153]. In general, higher eukaryotes have a “cap2” while lower eukaryotes have “cap 1” [12,154,155].

### 4.2. *CMTR1* and Diseases

*CMTR1* (*ISG95*, *FTSJD2*, *KIAA0082*) is the human 2′-O-methyltrasferase that modifies the N1 (first transcribed nucleotide) of the mRNA cap ([151] and Figure 3). Deregulation in *CMTR1* expression have been linked to different disease conditions, mostly through mechanisms related to distinguishing self from non-self RNA [156]. This is expected as the Nm-modified N1 of the cap serves as part of the innate host defense mechanism. In contrast, uncapped RNAs, including viral transcripts, trigger an interferon-mediated antiviral response [157,158,159,160,161].

### 4.3. *CMTR1* and Asthma

Inhaled corticosteroids (ICS) are the most effective controller medications for asthma, but some patients experience severe asthma exacerbations despite ICS treatment [162]. This variability in ICS response is due to genetic variations and the *rs2395672* SNP in *CMTR1* has been associated with increased risk of exacerbations. Moreover, *CMTR1* mRNA was overexpressed in nasal lavage samples from patients experiencing exacerbations in an independent microarray data set. These studies implicate CMTR1, an mRNA Nm methyltransferase, as a novel candidate gene with potential roles in the pathogenesis of asthma exacerbations [156]. The exact role of *CMTR1* in asthma is not clear but could be related to the fact that the transcription of human *CMTR1* is interferon-stimulated and *CMTR1* participates in cellular defense mechanisms against viral infection [163,164]. It should be noted that respiratory viruses are major triggers of exacerbations and a major cause of morbidity and mortality in adults and children with asthma [156,165]. Thus, by regulating mRNA stability and transcriptional expression of interferon-induced genes, *CMTR1* may play an important role in regulating genes involved in immune responses to viral infections.

### 4.4. *CMTR1* and Alzheimer’s Disease

Alzheimer’s disease (AD) is the most common form of dementia [166], characterized by the formation of amyloid plaques (Aβ). Chronic activation of microglial cells creates a pro-inflammatory environment, which is believed to be central for the development of the disease as well as its progression [167]. There is an upregulation of immune system-related pathways such as interferon-gamma signaling pathways at 10 weeks old (versus 6 weeks old) in the brain of AD mice model (5xFAD mice). Interestingly, in those mice, *CMTR1* was found among the 20 most upregulated proteins [168]. Again, we can imagine that *CMTR1* supports the expression of genes associated with this innate immune response, and thus contributes to the development of AD.

### 4.5. *CMTR1* and Cancer

*CMTR1* has not been directly linked to cancer, but multiple studies suggest a possible role in tumors. Microorganisms’ DNA/RNA signatures and integration sites in the host genome are promising biomarkers for human oral cavity and oropharyngeal squamous cell carcinomas (OCSCC/OPSCC) diagnosis and prognosis. Genomic elements of the oncogenic John Cunningham (JC) polyomavirus are found in the chromosomes of OCSCC cells and among the concerned integration sites are intronic regions of *CMTR* [169]. It is possible that these insertions alter *CMTR* gene expression in a way that promotes oncogenesis. It would be interesting to deepen that study by applying the RiboMethSeq approach or others Nm detection methods to highlight any potential loss of CMTR-dependent Nm modifications in the concerned cancer tissues.

Mutations in Janus kinase 3 (JAK3) occur frequently in T-cell acute lymphoblastic leukemia (T-ALL) and are able to drive cellular transformation and the development of T-ALL in mouse models [170]. JAK3 is a tyrosine kinase that can control protein activity and is an attractive therapeutic target to help establish a more patient-specific therapy [171]. Curiously, *CMTR1* phosphorylation is significantly increased upon inhibition of JAK3 activity [172]. Thus, *CMTR1* activity seems to be affected in a cancer-associated mutant context and may be one of the downstream explanations for the JAK3 mutant cancer phenotype. 

Finally, NSCLC (non-small cell lung cancer, already mentioned in the *FTSJ2* section) accounts for approximately 80–85% of lung cancers and is a leading cause of cancer-related mortality in both men and women worldwide [173,174]. *ALK* gene rearrangements, producing fusion oncogenes, is a driving mutation underlying the development of NSCLC [175]. The *EML4-ALK* oncogene is already well described [176,177], but *ALK* can also form rearrangements with *CMTR1*. It is yet to be confirmed if the *CMTR1-ALK* fusion gene is also oncogenic and if it has a role in cancer drug resistance [178]. 

### 4.6. mRNA and Splicing

The removal of introns from the precursor mRNA (pre-mRNA), or the so-called splicing, is an essential step in the production and diversity of functional mRNAs transcripts. The process is ensured by the spliceosome machinery that contains small nuclear ribonucleoprotein (snRNP) units (reviewed in Reference [179]).

### 4.7. snRNAs

The RNA components of the spliceosome are the small nuclear RNAs (snRNAs), most of which possess multiple modified nucleotides, including Nm and pseudouridine (ψ) positions that support the strong secondary structure and protein-binding capacity of snRNAs. Many of those modifications are essential for spliceosome assembly or efficiency (reviewed in References [180,181]. For example, the 5′- end of U2 snRNA contains four specific 2′-O-methylated bases and each of them is required for splicing [182]. Importantly, U2 snRNA seems to play a role in alternative splicing in the brain as disruption of one of the U2 copies causes cerebellar ataxia and neurodegeneration in mice [183]. It would be interesting to know if the loss of individual modifications on snRNAs can be responsible for any disease conditions. 

### 4.8. scaRNAs

The majority of modifications on snRNAs are applied in the Cajal bodies with the help of small Cajal body specific RNAs (scaRNAs) [184,185,186]. ScaRNAs are structurally and functionally indistinguishable from snoRNAs and can even interact with common enzymes like fibrillarin and dyskerin to guide the modifications on their target RNAs [187,188]. Their classification can be challenging but the conventional view is that they differ in their localization and type of targeted RNAs [189]. The role of scaRNAs in the maturation of spliceosomal snRNAs and the fact that alternative splicing is particularly prevalent in the brain suggest that neurons could be among the cell types most sensitive to disruption of scaRNA activity [190].

Alternative splicing also plays an important role in regulating mammalian heart development. The decreased expression of 12 scaRNAs was reported in congenitally abnormal human hearts with subsequent dysregulated alternative splicing of several genes associated with heart development [191]. Nine of the scaRNAs carry a C/D box and are predicted to participate in Nm modification of U2 and U6 snRNAs [191]. The phenotype seems to be conserved among vertebrates as a knock-down of the C/D box containing *SNORD94* produces developmental abnormalities, including heart malformations, which is also found in zebrafish [192].

### 4.9. mRNAs and Internal Nm Modifications

Modification studies on mRNAs have been mostly limited to the abundant and easily detectable m^6^A modification until the discovery that tRNA modifying enzymes like pseudouridine synthases (Pus) can also target mRNAs [193,194,195].

When it comes to Nm modifications, the tRNA 2′-O-methyltransferase, TRM44 (mentioned in previous chapter 2.4) was found to associate with polyadenylated (poly(A)) RNA in yeast, suggesting it may also target mRNAs in addition to tRNAs [196]. It is actually not uncommon for RNA-modifying enzymes to have multiple RNA targets [43] and even C/D box snoRNAs, typically reserved for rRNA Nm modification, that can target tRNAs in Archaea [197]. Interestingly, this capacity of a C/D box RNA to target tRNA instead of rRNA seems to be conserved in human (Vitali and Kiss, personal communication).

In yeast, Spb1, the rRNA 2′-O-methyltransferase (mentioned in a previous chapter), was found not only to associate with mRNAs, but also to methylate specific positions, methylations that seem to maintain their steady state level [198]. The human ortholog FTSJ3 has also been found in the poly(A) mRNP (messenger ribonucleoprotein) interactomes in multiple human cell lines [196,199,200], and its capacity to Nm-modify mRNAs awaits confirmation [55]. However, very recently, FTSJ3 was shown to methylate viral HIV-1 RNA [201]. This Nm deposition by FTSJ3 on HIV-1 RNA seems crucial for the virus to evade innate immune recognition. Although a recent discovery, mRNA internal (outside the cap) Nm modifications are officially a fact in mammalian mRNAs. They are mostly found in CDS, and especially near splice sites, but also in 5′ and 3′ UTR (untranslated regions), as well as in introns (suggesting that Nm is installed co-transcriptionally in the nucleus), and a small fraction in alternatively spliced regions [4]. Moreover, Nm within coding regions of mRNA disrupts key steps in codon reading during translation. More specifically, Nm on mRNA CDS leads to excessive rejection of cognate aminoacylated-tRNAs, probably due to sterical perturbation of the ribosomal-monitoring bases with cognate codon-anticodon helices interactions [202].

As it took time to find Nm modifications in mRNAs due to the low abundance of mRNAs compared to other classes of RNA, related diseases have not been described yet. Still, as more and more tRNA and rRNA 2′-O-methyltransferases seem to also target mRNAs [198], the diseases associated to these enzymes may also be due to the effects on their mRNA targets.

## 5. Nm in Small RNAs Silencing Pathways

The small non-coding RNA (sncRNA) silencing pathways regulate gene expression transcriptionally or post-transcriptionally. There are three main classes of sncRNAs: mi- (micro), si- (small-interfering), and piRNAs (PIWI-interacting RNAs). Each of them partners with an endonuclease from the argonaute (Ago) family and guides through sequence complementarity the silencing of a target mRNA [203,204]. In order to escape poly-uridilation and subsequent exonuclease degradation, miRNAs and siRNAs in plants, and siRNAs and piRNAs in animals need to receive a 2′-O-methylation on their 3′-end. 

### HENMT1

The Nm modification at the 3′- end of the concerned small RNAs is catalyzed by HENMT1 (HUA enhancer 1, also known as pimet for piRNA methylase in animals), and is particularly important for the PIWI/piRNA pathway that protects gonadal cells from transposable elements (TE) insertional mutagenesis in animals [6,10,205,206,207,208,209,210,211]. A recent paper investigating a Henmt1 mouse mutant demonstrated that HENMT1 methyltransferase is essential for mouse spermatogenesis [212]. Lack of HENMT1 and the associated reduction in methylation of the piRNA 3′- end led to piRNA instability and de-repression of TE in germ cells results in malformed spermatids and male infertility [212]. Consistent with HENMT1 having an important role in the human testis, mRNA *HENMT1* is expressed in all germ cell types and HENMT1 proteins localized in the cytoplasm of all stages of germ cells (spermatogonia, spermatocytes, and spermatids), as well as Sertoli cells, with highest levels in spermatocytes [213]. Moreover, a pathological fusion transcript between regulator of chromosome condensation 1 (*RCC1*) and *HENMT1* was identified in testicular germ cell tumors [214].

However, the ancestral role of the piRNA pathway does not seem to have been restricted only to the germline [215,216,217]. Small amounts of piRNAs have been found in epithelial cells in *Hydra* (Cnidaria phylum) [218] and seem to be common among the somatic lineage of the Arthropod phylum [219]. In *Aplysia* (Mollusca phylum), piRNAs were described in the central nervous system where they seem to play a role [220]. HENMT1 Nm methyltransferase being essential for piRNA stability in germ cells is also implicated in the somatic-piRNA related biological processes in *Drosophila melanogaster*. There is also evidence for somatic role of HENMT1 related to Ago2-loaded small RNAs, its other RNA targets in *D. melanogaster*. Indeed, fly mutants for *HEN1* (*pimet*) show an accelerated neurodegeneration (brain vacuolization) and memory default [221], which are phenotypes common for Ago2 mutants [221,222].

## 6. Conclusions

This review aimed to synthesize most of the Nm related human diseases known to date in a systematic manner. However, the structuring task was hampered as, often, the same enzymes can modify different classes of RNAs, like SPB1 that acts both on mRNA and rRNA for example. Also, sometimes, small guide RNAs need to Nm-modify other small guide RNAs that will only then regulate gene expression, like in the case of scaRNAs and snRNAs. One could justifiably feel jealous of the multitasking capabilities that Nm enzymes and guides can have and of the interesting interplay as well as the extraordinary conservation (enzyme and targets) that exists between them. 

In conclusion, Nm modification is a potent source for better disease understanding that has been shadowed for a long time by the difficulties in detecting it. With the latest technical advances toward Nm detection, we are awaiting more insights in the biological roles of the still mysterious and ancient 2′-O-methylation RNA modification. Finally, we want here to sincerely acknowledge all pioneers on RNA modifications discovery (mostly rRNA and tRNA at the time) that paved the road brick by brick for us in this re-emerging field of RNA modifications.

## Figures and Tables

**Figure 1 genes-10-00117-f001:**
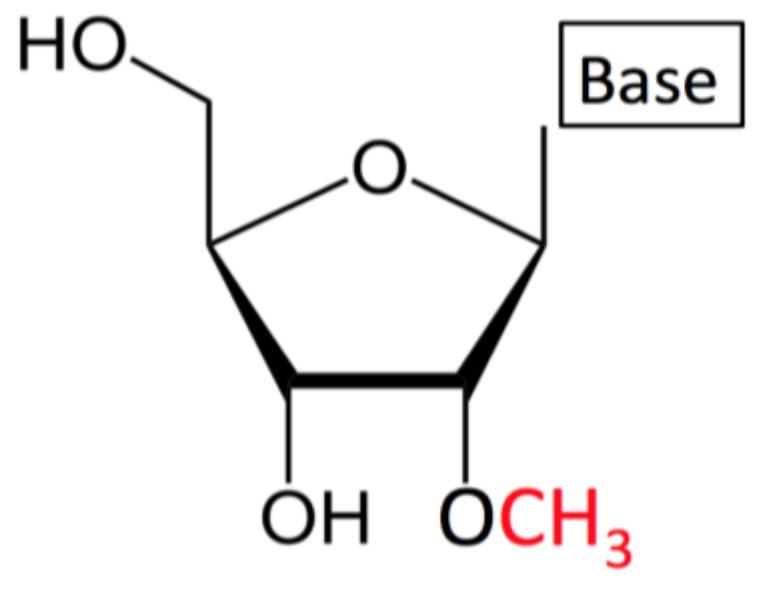
2′-O-methylated ribonucleoside (Nm): A –CH_3_ group is replacing the –H at the position 2′ in the ribose moiety. This modification can appear on any nucleotide regardless of the type of nitrogenous base (Base).

**Figure 2 genes-10-00117-f002:**
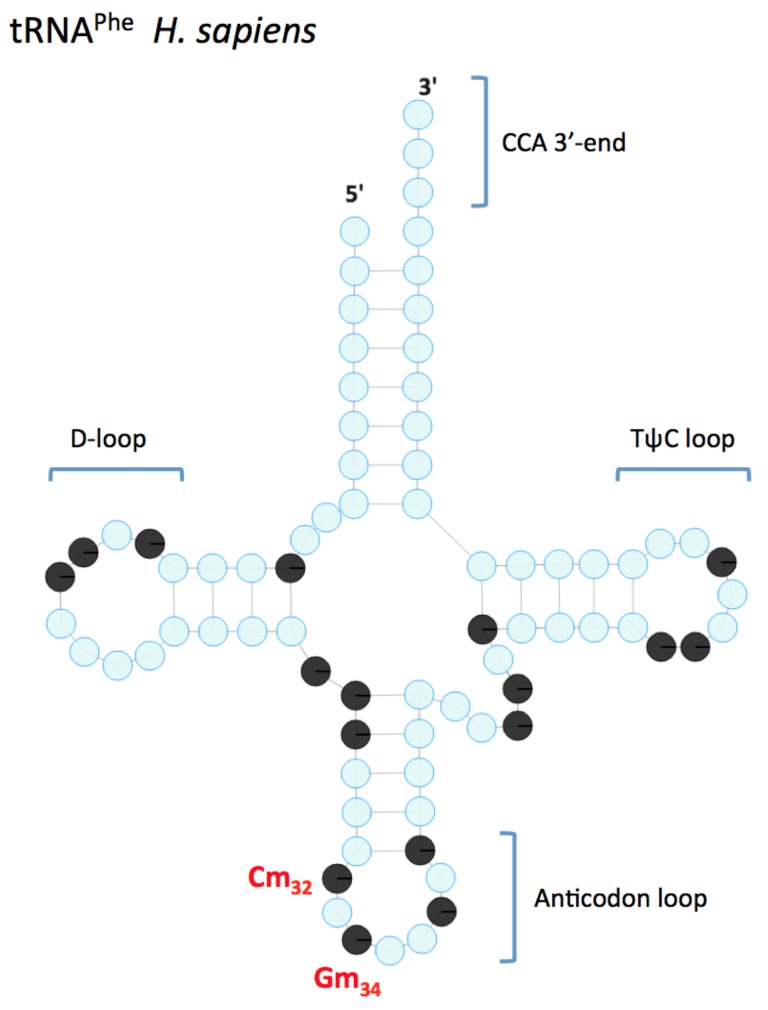
Representation of the mature human tRNA^Phe^ (76 nucleotides). The two Nm modifications in the anticodon loop, in C_32_ and G_34_, are placed by the 2′-O-methyltransferase *FTSJ1* and are annotated in red. Blue circles are for non-modified nucleotides, black circles mean modified nucleotides (adapted from http://modomics.genesilico.pl/).

**Figure 3 genes-10-00117-f003:**
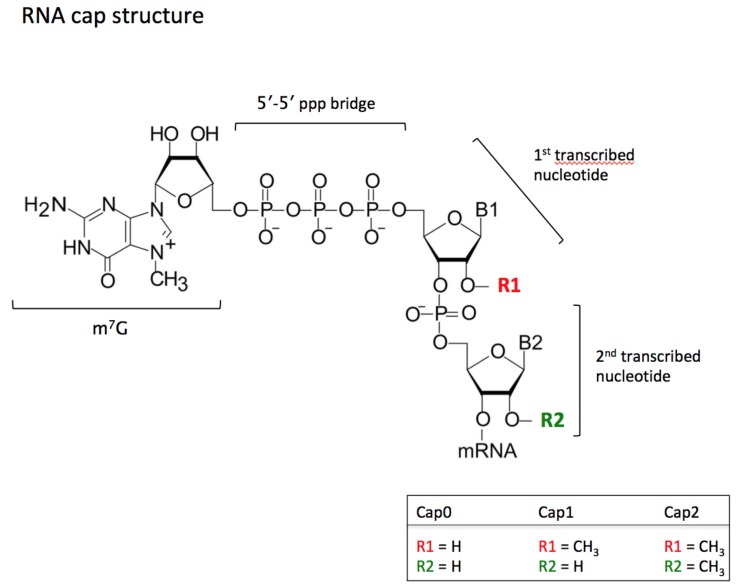
RNA cap structure: The cap0 structure consists of a guanosine residue, methylated in the N-7 position, which is bound to the terminal 5′-end nucleotide of the mRNA with a 5′-5′ triphosphate bridge. Subsequent 2′-O-methylation in the ribose of the first, or both the first and the second, transcribed mRNA nucleotides leads to the formation of cap1 or cap2, respectively. This figure has been adapted from Reference [151].

**Table 1 genes-10-00117-t001:** *FTSJ1* mutations associated with NSXLID (ss: splice site mutation, >: substitution, del: deletion, c.xxx: indicating the nucleotide (xxx) mutated on the gene coding DNA sequence (CDS), p.ZxY indicating that amino acid Z is changed by Y in the mutant and x indicates the AA position on the protein. HUS: Hôpitaux Universitaire de Strasbourg.

*FTSJ1* Allele	Family	Mutation/Location	Effect	Reference
Ftsj1Δ	6 (AU)	Deletion of *FTSJ1* and SLC38A5	Loss of FTSJ1	[61]
FTSJ1-ss	A3	c.121 + 1delG, p.Gly41Valfs*10 (IVS2, G DEL, + 1)/ Exon 2	Significant reduction of *FTSJ1* mRNA level (NMD)	[59]
196C > T	P48	c.196C>T, p.Gln66*/ Exon 4	Almost undetectable *FTSJ1* transcripts (NMD)	[59]
655G > A	MRX44	c.655G > A, p.Glu191_Tyr218del/ Exon 9	Loss of exon 9, protein lacking 28 amino acids	[59]
A > G	MRX9	c.192-2A>G, p.Gly65Cysfs*18 (IVS3AS, A-G, -2)/ Intron 3	Truncated protein	[60]
G > A	MRW06	c.571 + 1G > A, p.Glu191Glyfs*44/ Intron 8	Significant reduction of *FTSJ1* mRNA level (NMD)	[63]
p.A26P	7	c.76G > C; p.Ala26Pro/ Exon 2	Altered *FTSJ1* protein function	[64]
A > T	de novo variation	c.362-2A > T, p.?/ Intron 5 of trascrit NM_012280.3	Unknown (probable loss of exon 6 of transcript NM_012280.3 causing a frameshift)	Amélie Piton & Elise Schaefer, HUS, personal communication
Y > N	de novo variation	c.34T > A; p.Tyr12Asn/ Exon 2	Deposited as pathogenic	Ambry Genetics Clinvar NCBI [66]

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
