# Peer review of "RNA 2′-O-Methylation (Nm) Modification in Human Diseases"

_genes, 2019, doi:10.3390/genes10020117_

Round 1

Reviewer 1 Report

General Comments

From a general point of view is very interesting how also other RNA modifications involved/linked to human diseases (for example RNA editing) seems to have several common aspects with Nm. For example, absence of C-to-U editing in mice leads to hyperactivation of microglia which is very similar to what mentioned by the authors in chapter 4.4. This strongly suggests that epitrascriptomics play an essential role in maintaining the homeostasis in the brain; it also suggests that these modifications might be co-dependent to some extent.

Specific Comments

Line 154-155:

“Alternatively, another protein may partially compensate for the loss of FTSJ1, thereby preventing FTSJ1 mutations from causing more severe disorders”.

Do the authors know what other protein might complement?

Regarding chapter 3.3: Are there cancer treatments acting specifically against Fibrillarin? If yes, I think it would be interesting and useful to discuss them here.

Line 296-298: “Another study compared two lung adenocarcinoma cell sublines and demonstrated that FTSJ2 mRNA levels were lower in the more invasive one”.

This seems to be in disagreement with what said right before (sentence above this one), is it possible to speculate a bit about the presumed discrepancy?

Regarding chapter 4.5: Was RiboMethseq actually used for comparing the presence of Nm between cancer tissue and their healthy counterparts? It would be really interesting to see if there is a difference between distribution and amount of the modification in cancer.

Minor Comments

Line 58: point of “view”

Line 81: remove an extra space before “FTSJ1”.

Line 187: delete accent in “méthyltransferase”.

Line 275: a gene in human SMN should be in italics

Line 278: remove an extra space  before“frequently”

The entire chapter 4.7 is in italics, this should be fixed.

Line 476: remove an extra space  before "mysterious"

Author Response

We thank Reviewer#1 for his helpful comments and corrections. 

Please find attached our detailed response (pdf). 

Sincerely, 

Clément CARRE. 

Reviewer 2 Report

Genes-419353: RNA 2’-O-methylation (Nm) modification in Human Diseases

In this thorough review of Nm in human diseases, the authors cover a wide range of literature.  True to the title, the Review focuses on implications for human disease and does not attempt to give great details about the methodology surrounding the past and present study of Nm modifications. Overall, I think the review is insightful and thoughtfully organized.  I have only a few minor criticisms:

Line 53-54:  Another significant limitation of RiboMethseq is the requirement for very high RNA-seq depth, which can be trivial for a single abundant species (such as rRNA) but prohibitive for use in a complex setting (such as mRNA). It would also be worthy to note here the other chemistry for mapping Nm by RNA-seq, namely Nm-seq (ref #4) and RibOxi-seq (Zhu et al.).

Line 95: The statement regarding "10% prevalence of ID in male patients" is unclear. Perhaps you mean that for male patients with ID, 10% have an X-linked etiology? Please clarify and cite a source.

Lines 263-267: Are anti-fibrillarin antibodies "rarely seen in other diseases", or are they actually common? The authors give many examples of diseases and seem to be showing that >50% of patients with those disease have anti-FBL antibodies. That seems to be “common” in these diseases.

Why is section 4.7 all italicized?

Some additional minor grammatical suggestions (suggested changes are italicized):

-line 54: “RiboMethseq is that it cannot be applied on too small RNAs”

replace too small by short

-line 58: “transcriptome wide point of vue” -> point of view

-line 62: “most abundant modification” -> most abundant modifications

-line 63: “localization and by consequence” -> localization and as a consequence

-line 89: "insured" is not standard English for this context; "placed" would be better

-line 188: “The snoRNA serves as a guide thanks to its sequence complementarity” -> The snoRNA serves as a guide due to its sequence complementarity

-line 216: “We can resume that FBL is an essential nucleolar protein that catalyzes the…”  -> Therefore, FBL is an essential nucleolar protein that catalyzes the…

-line 258: “Those ANoAs doesn’t seem to affect” -> Those ANoAs don’t seem to affect

-line 416: “to target tRNA instead of rRNA looked conserved in human” - > to target tRNA instead of rRNA seems to be conserved in human

-line 435: “Nm in small RNAs silencing” -> Nm in small RNAs silencing pathways

Author Response

We thank Reviewer#2 for his helpful comments and corrections. 

Please find attached our detailed response (pdf). 

Sincerely, 

Clément CARRE. 
